# The Effects of Thermal Pasteurisation, Freeze-Drying, and Gamma-Irradiation on the Antibacterial Properties of Donor Human Milk

**DOI:** 10.3390/foods10092077

**Published:** 2021-09-02

**Authors:** Katherine Blackshaw, Jiadai Wu, Peter Valtchev, Edwin Lau, Richard B. Banati, Fariba Dehghani, Aaron Schindeler

**Affiliations:** 1School of Chemical and Biomolecular Engineering, Faculty of Engineering, The University of Sydney, Sydney, NSW 2006, Australia; kbla6530@uni.sydney.edu.au (K.B.); jiadai.wu@sydney.edu.au (J.W.); peter.valtchev@sydney.edu.au (P.V.); fariba.dehghani@sydney.edu.au (F.D.); aaron.schindeler@sydney.edu.au (A.S.); 2Centre for Advanced Food Engineering, The University of Sydney, Sydney, NSW 2006, Australia; 3Westmead Institute for Medical Research, Westmead, NSW 2145, Australia; edwin.lau@sydney.edu.au; 4Faculty of Medicine and Health, Medical Imaging Sciences, Brain and Mind Centre, University of Sydney, Camperdown, NSW 2006, Australia; 5Australian Nuclear Science and Technology Organisation, Locked Bag 2001, Kirrawee DC, NSW 2232, Australia; 6Mothers Milk Bank Charity (Human Milk Emergency Reserve-Project), P.O. Box 538, Tugun, QLD 4224, Australia; 7Bioengineering and Molecular Medicine Laboratory, The Children’s Hospital at Westmead and Westmead Institute for Medical Research, Westmead, NSW 2145, Australia

**Keywords:** gamma-irradiation, freeze-drying, donor human milk, antimicrobial, pasteurisation, milk bank, *Staphylococcus aureus*, *Salmonella typhimurium*, *Escherichia coli*

## Abstract

The most common pasteurisation method used by human milk banks is Holder pasteurisation. This involves thermal processing, which can denature important proteins and can potentially reduce the natural antimicrobial properties found in human milk. This study assesses the application of a hybrid method comprised of freeze-drying followed by low-dose gamma-irradiation for nonthermal donor human milk pasteurisation. Freeze-drying donor human milk followed by gamma-irradiation at 2 kGy was as efficient as Holder pasteurisation in the reduction of bacterial inoculants of *Staphylococcus aureus* (10^6^ cfu/mL) and *Salmonella typhimurium* (10^6^ cfu/mL) in growth inhibition assays. These assays also demonstrated that human milk naturally inhibits the growth of bacterial inoculants *S. aureus, S. typhimurium*, and *Escherichia coli.* Freeze drying (without gamma-irradiation) did not significantly reduce this natural growth inhibition. By contrast, Holder pasteurisation significantly reduced the milk’s natural antimicrobial effect on *S. aureus* growth after 6 h (−19.8% *p* = 0.01). Freeze-dried and then gamma-irradiated donor human milk showed a strong antimicrobial effect across a dose range of 2–50 kGy, with only a minimal growth of *S. aureus* observed after 6 h incubation. Thus, a hybrid method of freeze-drying followed by 2 kGy of gamma-irradiation preserves antimicrobial properties and enables bulk pasteurisation within sealed packaging of powderised donor human milk. This work forwards a goal of increasing shelf life and simplifying storage and transportation, while also preserving functionality and antimicrobial properties.

## 1. Introduction

Infants are born with an immature immune system and, therefore, rely on immunity acquired through breastfeeding to provide an infant <6 months with adequate nutritional and immune support. Where breastfeeding is not possible, the supply of donor human milk (DHM) is associated with lower morbidity and mortality rates than seen with infant formula [1,2,3]. A greater supply of DHM would, thus, be the best way to reduce infant morbidity and mortality rates [1]. However, there is currently a greater demand for DHM than supply [4,5,6]. Inefficient and unoptimized processing and storage methods are a major contributing factor to this shortage.

Almost universally, DHM is processed using Holder pasteurisation to ensure microbial safety and is stored at −20 °C until use. Holder pasteurisation, which involves heating the milk to 62.5 °C for 30 min, is currently limited to small-batch processing and is associated with the loss of antimicrobial properties [7,8,9,10]. Additionally, the loss of bioactive components and native milk bacteria may make the milk more susceptible to contamination [11]. The storage of milk is limited to 3–6 months depending on the country-specific guidelines [5,6].

Freeze-drying and gamma-irradiation may offer advantages over current processing, including a comparably extended shelf-life, as well as other advantages, such as reduced storage costs, larger capacity for storage, the possibility of upscaling, and international distribution. Freeze-drying may be a more favourable powderisation option than spray-drying as it is a nonthermal drying method. Because spray-drying uses streams of hot gas up to 200 °C to evaporate the water content, there are issues with the loss of protein function and stability, loss of fat content due to wall deposition [12], and decreased solubility of the powdered product due to association between casein micelles [13]. Freeze-drying has less impact on the natural bioactivity of human milk functionality and better yield and recovery [14,15]. For these reasons, freeze-drying is the preferred drying process for live-attenuated vaccines [16]. Consequently, several studies have now reported on the effects of freeze-drying on DHM. No significant effect has been found on the protein profile, the oligosaccharides, or the fatty acid profile in human milk after freeze-drying [17,18,19]. Significantly, freeze-drying retains the lactoferrin content of human milk, but a decrease in lysozyme and antioxidants has been reported [19].

Gamma-irradiation may be a suitable nonthermal pasteurisation process for powdered DHM as it can be applied to a powdered product in terminal packaging, while retaining bioactivity. Gamma-irradiation is a well-understood and widely used technology for preservation and pathogen control of food [20,21]. Doses of 1–10 kGy are used currently for pathogen reduction for fresh fruit, fresh and frozen meat, and dehydrated vegetables. Higher doses (10–50 kGy) are used for the sterilisation of meat and pathogen reduction of spices [21]. It has not yet been tested for powdered milk. The main mechanism of action is due to the direct radiolysis products of water, which are hydroxyl radicals and hydrogen peroxide, destroying parts of the DNA molecules in microorganisms [21]. However, irradiation may also induce changes to the food molecules, particularly at higher doses. This can include fat oxidation by oxygen radicals, damage to protein structures due to breaks in sulphur-containing amino acids, and a reduction in vitamins C and E [21]. The use of gamma-irradiation has been more extensively studied for food safety and quality than any other processing technology [21]. It is safe for use on food even at higher doses (>10 kGy); however, doses above 10 kGy may suffer from sensory changes [22].

This paper first assesses the suitability of gamma-irradiation as a pathogen control process for freeze-dried DHM (FD-DHM). Both intrinsic contamination of the milk (prior to freeze-drying) and a post-freeze-drying contamination scenario were considered. Secondly, the antimicrobial activity of gamma-irradiated FD-DHM at several different doses (2, 5, 10, 15, 25, and 50 kGy) were compared to raw DHM, FD-DHM, Holder-pasteurised DHM, and infant formula. In addition, the use of standard Holder pasteurisation on FD-DHM was also investigated. *Staphylococcus aureus, Salmonella typhimurium*, and *Escherichia coli* were selected as test pathogens as they are well characterised and commonly associated with sudden infant death syndrome [23], food poisoning [24], and contamination in infant formula [25].

## 2. Materials and Methods

### 2.1. Materials

LB broth and nutrient agar were made up from Bacto^TM^ Tryptone Pancreatic Digest of Casein and Bacto^TM^ Yeast Extract, purchased from Bacto Laboratories Pty, Ltd., (Mt Pritchard, NSW 2170, Australia), and NaCl and Agar A bacteriology Grade purchased from Astral Scientific (Taren Point, NSW 2229, Australia). Selective agars Xylose Lysine Deoxycholate and Eosin Methylene Blue were purchased premade from Edwards (Narellan, NSW, Australia). *Staphylococcus aureus* ATCC 12600^TM^*, Salmonella typhimurium* ATCC 14028^TM^*,* and *Escherichia coli* ATCC 43888^TM^ were purchased from Cell Biosciences (Victoria, Australia). STYO 9 was purchased from Thermo Fisher Scientific (Waltham, MA, USA). DAPI was purchased from Sigma-Aldrich (St. Louis, MO, USA).

### 2.2. Donor Human Milk Handling

Frozen raw DHM from healthy blood-tested donors was provided by Mothers Milk Bank Charity with written consent from the donors. Samples from multiple donors (*n* = 5–6) were thawed, pooled, and aliquoted into Eppendorf tubes and stored at −80 °C. Multiple, different pools of DHM were used for this study; however, all pools were treated and stored under the same conditions to minimize variation.

### 2.3. Freeze-Drying

Pooled DHM was aliquoted into 4 mL clear glass vials and frozen at −80 °C and then freeze-dried for 24 h in a general-purpose tabletop freeze dryer (Martin Christ GmbH, Osterode am Harz, Germany). The main freeze-drying phase was held at −75 °C at 3 mbar. The final freeze-drying phase was held at −8 °C at 0.0010 mbar. The overall procedure was 48 h. Freeze-dried DHM samples were kept at −80 °C until analysis.

### 2.4. Holder Pasteurisation

Pooled raw DHM and reconstituted FD-DHM were heated in Eppendorf tubes in quantities of 2 mL in a ThermoScientific Reacti-therm III TS 18823 heater to 62.5 °C (+/− 0.5 °C), held for 30 min once an internal temperature of 62.5 °C had been reached, followed by immediately cooling on ice. Samples were stored at −80 °C until analysis.

### 2.5. Gamma-Irradiation

Freeze-dried milk was irradiated in a cobalt-60 Gammacell irradiator at the ANSTO Gamma Technology Research Irradiator (GATRI) facility. Samples were kept at room temperature and irradiated with a dose rate of 0.78 kGy per hour and cumulative doses of 2 kGy, 5 kGy, 10 kGy, 25 kGy and 50 kGy. Absorbed irradiation doses were confirmed by ferrous sulphate dosimetry.

### 2.6. Reconstitution of Freeze-Dried Donor Human Milk

In total, 125 mg samples of freeze-dried, and freeze-dried and irradiated DHM were reconstituted in 1.5 or 2 mL Eppendorf tubes with 875 μL of sterile and de-ionized Milli-Q^®^ water heated to 40 °C and vortexed until the milk powder had dissolved.

### 2.7. Bacterial Stock Preparation

Bacterial stocks were grown overnight in LB broth at 37 °C from glycerol stock stored at −80 °C. *S. aureus* was cultured on nutrient (LB) agar. *S. typhimurium* and *E. coli* were cultured on selective agars (Xylose Lysine Deoxycholate agar and Eosin Methylene Blue agar, respectively). For each bacterial strain, a single colony was picked and grown overnight at 37 °C in LB broth. A total of 100 µL of the inoculated broth was pipetted into saline (0.9% NaCl), and the optical density at 600 nm was measured using a Cary 300 Bio UV-Visible Spectrophotometer (Agilent Technologies, Victoria, Australia). The optical density was adjusted to a bacterial load of 10^8^ cfu/mL.

### 2.8. Gamma-Irradiation of Freeze-Dried Milk for Pathogen Control

A concentration of 10^5^–10^6^ cfu/mL was prepared by addition of 100 µL of *S. aureus* and *S. typhimurium* bacterial stock to 920 µL of raw DHM in 1.5 mL tubes. A baseline measurement was performed after a serial dilution to the appropriate bacterial concentration for each raw DHM sample for viable colony detection. Samples were then transferred on ice for freeze-drying and gamma-irradiation. After freeze-drying, samples were gamma-irradiated within 1 week for *S. aureus* and 2 weeks for *S. typhimurium*. Holder pasteurisation of reconstituted FD-DHM was performed within one week of freeze-drying treatment. Samples were stored at 4 °C at all times. Plate assays were performed by spreading 10 µL of milk sample onto nutrient agar plates divided in half, which were incubated at 37 °C for 24 h. Each sample was analysed at least in duplicate. A further test of pathogen revival during storage was performed for all treatments by storing all reconstituted samples at −80 °C for 2–3 weeks and then spreading 50 µL of undiluted sample onto nutrient agar plates. Control groups included inoculated LB broth and uninoculated raw DHM.

### 2.9. Bacterial Proliferation Assays

Antimicrobial activity was analysed by measuring the growth inhibition of inoculated pathogens compared to an optimum growth model. A bacterial inoculation of 10^7^ cfu/mL was achieved by adding 100 µL of bacterial suspension to 920 µL of each milk groups being tested and to 920 µL of LB broth. The latter was used as a positive control for the test pathogen growth. This inoculation was used for all test pathogens and all experiments except where noted. A 20 µL quantity was taken from each sample prior to inoculation as a negative control. After inoculation, all samples were incubated at 37 °C and stirred at 200 rpm. Growth of viable colonies was analysed by serial dilution to appropriate concentration and spreading 10 µL of sample onto nutrient agar plates divided in half at designated time points (either 0, 2, 4, and 6 h or 0, 6, and 24 h). Nutrient agar plates were incubated overnight at 37 °C, and viable colonies were counted. For all experiments, 2 technical replicates were performed for colony counts and expressed as cfu/mL. For measurement of antimicrobial properties of raw DHM, HoP, FD, FD+HoP, and irradiated FD-DHM biological replicates were performed, and data of the biological replicates were collated into one dataset. This model of analysis was designed to account for the natural variation that occurs with human milk components, and the milk quality due to storage conditions and handling prior to collection. The inhibition of bacterial growth was expressed as a percentage compared to the growth curve of the inoculants in a positive control (LB broth), as shown by the formula below:$$\mathrm{Percentage}\;\mathrm{of}\;\mathrm{growth}\;\mathrm{inhibition}=100-\left(\frac{\mathrm{colony}\;\mathrm{count}\;\mathrm{of}\;\mathrm{sample}\;}{\mathrm{colony}\;\mathrm{count}\;\mathrm{of}\;\mathrm{LB}}\times 100\right)$$

### 2.10. Preparation of Bacteria for Growth Curve Analysis and Flow Cytometry

Flow cytometry was used to measure the presence of live and dead bacteria. Bacterial stock was prepared as stated above. However, a smaller bacterial inoculation of 10^6^ cfu/mL was used, which was achieved by adding 150 µL from the bacterial suspension to 1350 µL of each of the DHM samples. A 20 µL quantity was taken from each sample prior to inoculation as a negative control. After inoculation, all samples were incubated at 37 °C and rotated at 200 rpm. At the timepoints of 0 and 24 h, 500 µL of each sample was transferred to a 15 mL Falcon tube for flow cytometry preparation. A serial dilution was performed at these time points for detection of viable colonies.

### 2.11. Flow Cytometry Assay Preparation

Bacteria were stained using a combination of SYTO 9 and DAPI for the detection of bacterial viability (live and dead) by flow cytometry. The standard staining procedure was used based on the protocol reported in the previous studies [26,27]. All DHM samples were centrifuged at 3000× *g* for 5 min to pellet bacteria. Pellets were triple-washed with ~10 mL saline at 3000× *g* for 5 min. After the third wash, the supernatant was removed, and the pellet was resuspended in 300 µL of saline. Samples were then stained with SYTO9 (Thermo Fisher Scientific, Waltham, MA, USA) and DAPI (Sigma-Aldrich, St. Louis, MO, USA) at a final concentration of 1.67 µM and 15 ug/mL respectively for 15 min on ice. After staining, samples were tested on the BD FACSCanto II Cell Analyzer (BD Biosciences, La Jolla, CA, USA). Sizing beads (0.88 and 1.34 um) were used to help locate *S. aureus* on FSC vs. SSC. Data acquisition was set to record a minimum of 10,000 *S. aureus* events or until sample ran out. Recorded data were analysed using FlowJo 10.7.0 for Windows (BD Biosciences, La Jolla, CA, USA). A single biological replicate was used for this experiment.

A standard curve for *S. aureus* was prepared by mixing 0%, 10%, 25%, 50%, 75%, and 100% 1 × 10^8^ cfu/mL of live *S. aureus* with the remaining percentage made up with dead suspensions of *S. aureus*. The suspension of dead *S. aureus* was originally taken from the same stock as the live bacteria and was boiled for 10 min before storing on ice.

In the control sample of *S. aureus* in saline, *S. aureus* populations were readily identified in the FSC vs. SSC plot followed by gating to SYTO9 staining using the blue laser at 488 nm. Live vs. dead bacteria populations suspended in saline can be identified by DAPI using the violet laser at 405 nm. A different gating strategy was developed for detecting live/dead bacteria in DHM by using an additional gate plotting from detectors B530 vs. B585. B530 detects STYO9, and B585 is an unused detector that was used to gate out the background fluorescence of the milk to improve detection of the bacteria. Bacterial concentration and flow rate were optimised to maximise singlet events. Positive/negative events of *S. aureus* in LB broth are shown in Appendix A. Detection of live/dead bacteria in raw DHM is shown in Appendix A.

### 2.12. Statistical Analysis

Changes in antimicrobial activity in treated DHM samples compared the antimicrobial activity in raw DHM and tested for significance using two-tailed *t*-tests in Excel (2016). GraphPad Prism 8.3.1 was used to create the figures.

## 3. Results

### 3.1. Efficacy of Pathogen Control Methods Prior to Freeze-Drying of Donor Human Milk

Liquid donor human milk samples were inoculated with a high dose (10^6^ cfu/mL) of *S. aureus* and *S. typhimurium*. All samples were then freeze-dried. The powderised DHM was then either gamma-irradiated (2–50 kGy) or Holder-pasteurised after reconstitution with sterile water.

Freezing at −80 °C and short-term holding for 24 h of the inoculated samples of raw DHM alone reduced the presence of the inoculated bacteria to 3.4 × 10^5^ cfu/mL while freeze-drying of the inoculated raw DHM led to a small log reduction of 0.9 for the inoculant *S. aureus* (6.1 × 10^4^ cfu/mL) and 0.5 (3.5 × 10^5^ cfu/mL) for *S. typhimurium*.

The results in Figure 1 show that the irradiation of FD-DHM at 2 kGy reduced the colony count to 50 cfu/mL for *S. aureus* and 1.5 × 10^2^ cfu/mL for *S. typhimurium*, while no viable colonies of either *S. aureus* or *S. typhimurium* were detected in any inoculated samples, which had been freeze-dried and gamma-irradiated at 5 kGy doses and above. Holder pasteurisation after reconstitution reduced the colony counts for *S. aureus* inoculated milk to 3 × 10^2^ cfu/mL and 50 cfu/mL for *S. typhimurium*. Thus, 2 kGy gamma-irradiation achieved a reduced bacterial inoculant reduction similar to Holder pasteurisation, while a 5 kGy dose and above removed all growth of the bacterial inoculant.

### 3.2. Efficacy of Gamma-Irradiation in Reducing Bacterial Inoculants Added to Freeze-Dried Donor Human Milk Powder

Extrinsic contamination of FD-DHM powder was simulated by inoculating 125 mg of non-reconstituted DHM powder with 10^6^ cfu/mL of *S. aureus* and *S. typhimurium* (bacteria tested separately). After the reconstitution of contaminated (but un-irradiated) FD-DHM samples, an immediate reading of 1.5 × 10^5^ cfu/mL of viable *S. aureus* and 2.5 × 10^5^ cfu/mL of *S. typhimurium* was seen. When inoculated FD-DHM powder was kept for one week at room temperature storage and then reconstituted, only 2.8 × 10^4^ cfu/mL of viable *S. aureus* and 3 × 10^2^ cfu/mL of *S. typhimurium* colonies were recovered; i.e., a log reduction of 1.1 for *S. aureus* and 2.6 for *S. typhimurium* had occurred, indicating that the viability of the inoculant bacteria in dry FD-DHM powder reduced over time. These values were presumed to be representative of all inoculated FD-DHM samples. When the inoculated un-irradiated FD-DHM, which had been stored for one week at room temperature, was reconstituted and additionally Holder-pasteurised, no growth in *S. aureus* was found, while for *S. typhimurium*, a log reduction of 3.4 (10^2^ cfu/mL) was seen (Figure 1). By contrast, gamma-irradiation at 2 kGy and above of *S. aureus*- and *S. typhimurium*-inoculated FD-DHM obliterated any subsequent bacterial growth, both after immediate reconstitution and 1 week storage, indicating that low-dose gamma-irradiation of FD-DHM powder further reduces the viability of the inoculant bacteria in dry FD-DHM powder.

### 3.3. Growth Inhibition of Pathogens by Raw Donor Human Milk

Donor human milk has intrinsic antimicrobial bioactivity, which is found in the macromolecules and immune components, such as lactoferrin, lysozyme, immunoglobulins, and κ-casein [28]. The expected antimicrobial activity of raw DHM was confirmed with an initial analysis of growth curves of pathogens *S. aureus*, *S. typhimurium*, and *E. coli*. All three pathogens were inoculated in separate raw DHM samples and incubated at 37 °C over 24 h. At an inoculation of 10^7^ cfu/mL for all three test pathogens, DHM inhibited the growth of *S. aureus* > *E. coli* > *S. typhimurium* (Figure 2). After 6 h incubation, DHM inhibited 99–100% of the growth of *S. aureus*, 41.6–41.8% of the growth of *S. typhimurium*, and 25.2–41.1% of the growth of *E. coli* compared to control growth curves (growth of inoculated bacteria in LB broth). After 24 h incubation, DHM inhibited 100% of the growth of *S. aureus*, 63.0–82.1% of the growth of *S. typhimurium*, and 59.9–71.6% of the growth of *E. coli* compared to control growth curves (Figure 2). Different levels of growth inhibition may reflect variation in infectious dose for each inoculant. On the other hand, the greater growth inhibition observed for *S. aureus* compared to the other Gram-negative inoculants may reflect antibacterial mechanisms in DHM having a greater specificity against *S. aureus* than *S. typhimurium* or *E. coli*. For example, a study on the antibacterial effects of lactoferrin found a higher growth inhibition of Gram-positive bacteria compared to Gram-negative bacteria [29]. Only one other study on growth inhibition of pathogenic bacteria in DHM was found, which did not find any difference between the inhibition of Gram-positive and Gram-negative bacteria [11].

### 3.4. Freeze-Drying Does Affect Antimicrobial Activity in DHM 

As shown in Figure 3, raw DHM caused a 97.0% ± 3.1 reduction in *S. aureus* at 6 h and 100% at 24 h, suggesting a potent antimicrobial bioactivity. Holder pasteurisation of DHM decreased growth inhibition of *S. aureus* to 76.2% ± 0.01 at 6 h and 97.6% ± 2.9 at 24 h. After freeze-drying, a similar amount of growth inhibition to raw DHM of 95.6% ± 2.9 at 6 h and 99.9% ± 0.1 at 24 h was observed. Holder pasteurisation significantly reduced the growth inhibition of DHM compared to the unprocessed control at 6 h by 19.8%, *p* = 0.01. However, this difference is largely lost by 24 h with only a 2.4% decrease in growth inhibition of inoculant in Holder-pasteurised DHM. While this is a small decrease when compared to the bacterial growth in the positive control (LB broth), 2.4% is equal to a 6-fold increase in bacterial growth (Figure 3). By contrast, freeze-drying negligibly affected the growth inhibition of DHM at all time points (Figure 3). It is important to note that raw DHM contains native bacteria such as *Bifidobacteriacae* that resist antimicrobial factors [30,31]. Bacteria were consistently observed in the non-inoculated control plates for raw DHM across different pools of DHM. These bacteria, which were identified as different to the inoculant by microscopy and colony morphology, were observed in all raw DHM samples at all timepoints and on some Holder-pasteurised samples at the 24 h timepoint. These bacteria have a distinctive morphology that can be clearly distinguished from *S. aureus* (Appendix A).

We investigated the effects of Holder pasteurisation on the antimicrobial properties of reconstituted FD-DHM. Infant formula, which is the current alternative on the market and also a powdered product, was also included by way of comparison. The infant formula was reconstituted using 45 °C sterile water and was not treated with any pathogen control method. At 6 h, the growth inhibition for infant formula was 91.4% ± 2.0, and after 24 h was 96.7% ± 0.5 (Figure 4a). At 6 h, growth inhibition for Holder-pasteurised FD-DHM was 90.7% ± 2.2 and after 24 h was 98.6% ± 0.24 (Figure 4a). Overall, Holder pasteurisation of FD-DHM demonstrated a similar amount of growth inhibition to infant formula, in contrast to gamma-irradiated milk, which demonstrated an increase in antibacterial properties.

### 3.5. Growth Inhibition of Gram-Negative Bacteria

Raw DHM was less effective at inhibiting the growth of *S. typhimurium* than S aureus, showing a 41.7% ± 0.1 reduction of *S. typhimurium* at 6 h and 72.5% ± 13.6 at 24 h. Holder pasteurisation of DHM transiently increased the growth inhibition of *S. typhimurium* to 57.0% ± 4.8 at 6 h, but by 24 h, the growth inhibition reduced to 48.6% ± 1.4, and thus the remaining antimicrobial effect of thermally treated milk was less than in raw DHM. After freeze-drying, the amount of growth inhibition was similar to raw DHM with a reduction of 38.2% ± 14.5 at 6 h and 86.7% ± 8.6 at 24 h (Figure 3).

Raw DHM showed a similar amount of reduction of *E. coli* growth as *S. typhimurium*. There was a reduction of 33.2% ± 11.2 at 6 h and 65.8% ± 8.3 at 24 h (Figure 3). After freeze-drying, very similar rates of reduction were found in raw DHM of 40.1% at 6 h and 64% at 24 h (Figure 3). With a lower inoculation of 10^5^ cfu/mL of *E. coli*, after 6 h, raw DHM inhibited growth of *E. coli* significantly more than Holder-pasteurised DHM (*p* = 0.02). By contrast, the impact of freeze-drying was negligible at 6 h (Figure 4b). However, after 24 h, in Holder-pasteurised DHM and in raw DHM samples, only very little detectable E coli colonies were observed on the agar plate, and bacteria with different colony morphology from *E. coli* were found. The colony morphology matched the colony morphology of the bacteria grown in the negative control agar plate for uninoculated raw DHM. In FD-DHM at 24 h, only the growth of *E. coli* was observed (Appendix A).

### 3.6. Flow Cytometry Results

Donor human milk has potent antimicrobial activity [32]. In growth inhibition assays of raw DHM, a 100% bacterial death rate of the inoculant *S. aureus* was observed after 24 h. To investigate further, we used flow cytometry and a live/dead stain to differentiate between live and dead bacteria in the test inoculants with *S. aureus* in raw DHM, Holder-pasteurised DHM, FD-DHM, and reconstituted FD-DHM that was Holder-pasteurised and infant formula. At baseline (0 h), the majority of bacteria in all samples were alive. After 24 h incubation at 37 °C, FD-DHM contained mostly dead bacteria (88.7% dead) in contrast to raw DHM and Holder-pasteurised DHM (7.8% and 6.1% dead bacteria, respectively). In reconstituted FD-DHM that was Holder-pasteurised, 99.7% of bacteria were dead after 24 h incubation at 37 °C. By contrast, for infant formula only 4.5% of bacteria were dead after 24 h (Table 1). The full datasets are shown in Appendix A (Appendix A).

It is likely that bacteria of the native milk microbiome in raw DHM samples contributed to the count of live bacteria detected by flow cytometry (after 24 h) since in the same raw DHM samples without added bacterial inoculant growth assays regularly showed bacterial growth of native bacteria.

### 3.7. Gamma-Irradiation Increases the Antimicrobial Activity of FD-DHM

To date, there has been minimal research on the effects of pathogen control measures on FD-DHM. The effect on the antimicrobial properties of gamma-irradiation on FD-DHM was analysed by measuring the growth inhibition of *S. aureus* in gamma-irradiated FD-DHM and compared to non-irradiated FD-DHM and raw DHM. Samples were inoculated with 10^6^ cfu/mL and 10^7^ cfu/mL of *S. aureus* and measured at 2, 4, 6, and 24 h. The effect of gamma-irradiation across 2–50 kGy on the preservation of antimicrobial properties in reconstituted DHM was compared (Table 2, Figure 5). For reconstituted DHM powder samples that had received 2 kGy gamma-irradiation, the growth inhibition over 6 h for *S. aureus* was 99.7% ± 0.3 (10^6^ cfu/mL) and 93.8% (10^7^ cfu/mL) with no growth after 24 h. In reconstituted samples of powder that had received ≥5 kGy gamma-irradiation no growth was seen with an inoculant dose of 10^6^ cfu/mL at either 6 or 24 h. With a ten-fold higher bacterial inoculant load of *S. aureus* (10^7^ cfu/mL) and a 6 h incubation, the growth inhibition exerted by the 5 kGy gamma-irradiated powder was 95.8% (Figure 5). In DHM powder irradiated at 25 and 50 kGy, the number of viable *S. aureus* colonies detected remained static over the first 4 h of incubation before the reduction in bacterial cell number was observed at 6 h (Figure 5).

## 4. Discussion

Our study demonstrates that a hybrid method comprised of freeze-drying followed by gamma-irradiation can be considered as a viable method for the pasteurisation and shelf-life extension of human milk with limited impact on its antimicrobial properties. The current regulations for the handling of the DHM state that any batch with a microbial count of 10 cfu/mL or more must be discarded [33]. For *S. aureus* and *S. typhimurium*, a 2 kGy dose of gamma-irradiation was sufficient to destroy a high dose of inoculated pathogenic strains when tested in reconstituted human milk powder samples. Only after the additional storage of reconstituted sample, which would be against the recommended practice of immediate consumption, a small number of viable bacteria were found in these samples. Therefore, a higher dose of 5 kGy might be recommended for the pasteurisation of FD-DHM. Previous studies have reported that doses between 0.2–0.8 kGy are sufficient to inactivate bacteria [34]. While gamma-irradiation has not been previously studied in the context of human milk, it has been assessed for bovine colostrum [35] and infant formula [36]. A dose of 2 kGy was sufficient for the elimination of all viable cultures of *E. coli* and *S. aureus* in bovine colostrum after spray-drying [35]. A higher irradiation dose of 5 kGy was necessary to eliminate all viable cultures in infant formula inoculated with 10^8^–10^9^ cfu/mL of *Cronobacter sakazakii* [36]. The pasteurisation by gamma-irradiation is mainly attributed to the damage to the DNA of bacteria or secondary radiolytic products through the interaction of radiation with water [34]. Thus, food products with low water content or in subzero temperatures may need higher irradiation doses.

The results of this study showing an antimicrobial activity for raw DHM is consistent with prior studies [37,38]. Holder pasteurisation reduced antimicrobial properties, which is consistent with thermal processing causing a loss of immune functionality [11,39]. It was also supported by previously reported data of diminished growth inhibition of *S. aureus* and *E. coli* in Holder-pasteurised DHM compared to raw DHM [11]. In our study, no significant difference was found in the growth inhibition of *S. aureus*, *E. coli*, or *S. typhimurium* in DHM after freeze-drying. Previous studies of freeze-dried DHM have shown that the bioactivities of key antibacterial components are retained more efficiently than after heat treatment [19,40].

Earlier studies have demonstrated the efficacy of gamma-irradiation using doses of between 10 and 40 kGy to inhibit long-term bacterial growth in dairy-like products [41] and ice cream [42]. The effectiveness of growth inhibition seen in the inhibition assays with our irradiated FD-DHM samples was in contrast to Holder-pasteurised DHM as well as reconstituted FD-DHM powder that had additionally been Holder-pasteurised; in both of which, the thermal treatment caused a loss of antimicrobial properties (Figure 4a). It is possible that bacterial inoculants did not grow in the reconstituted irradiated FD-DHM powder due to the minimal impact of freeze-drying [40] and gamma-irradiation on the bile-salt-stimulated lipase activity and, hence, its preservation in the reconstituted samples [43]. Lipase increases lipid hydrolysis, thus increasing free fatty acids and monoglycerides, both known to have strong antimicrobial properties [44,45,46,47,48,49,50,51,52,53,54]. More recently, we have found an elevated level of free fatty acids in the volatiles of irradiated samples compared to raw and Holder-pasteurised DHM (unpublished data). Additionally, previous studies also show that gamma-irradiation of other food products with high lipid content have dose-dependent increases in free fatty acids and decreases in unsaturated fatty acid content [55]. Fatty acid and lipid oxidation products, as they occur during rancidification in food, are traditionally known to allow long-term storage of fat rich substances due to their antimicrobial properties.

By contrast, bile salt-stimulated lipase in DHM is known to be destroyed by Holder pasteurisation [56,57]. Reduced lipid hydrolysis might thus be a factor in the reduced overall growth inhibition observed in Holder-pasteurised DHM and in Holder-pasteurised FD-DHM.

It was noted in this study that the natural milk bacteria may play an important role in inhibiting pathogenic growth. Human milk is known to have natural populations of different types of bacteria [30]. Previous studies have found that certain bacteria in milk specifically inhibit pathogens [58]. Although testing for natural milk bacteria was beyond the scope of this study, it is most likely that the observed bacteria in negative control agar plates for raw DHM prior to inoculation with a pathogen were natural milk bacteria (Appendix A). These bacteria may have contributed to the antibacterial properties of the untreated raw milk.

## 5. Conclusions

This study demonstrates the improved retention of antimicrobial properties of FD-DHM and irradiated FD-DHM compared to Holder-pasteurised DHM. Gamma-irradiation of 5 kGy is sufficient to reliably eliminate all bacteria in DHM. The hybrid technique developed in this study might open an avenue for bulk processing of DHM stored in milk banks. This technique enables in-packaging pasteurisation of dried DHM powder with a longer shelf-life and more convenient storage, transportation, and accessibility for newborn babies in hospitals or other places in demand, such as crisis conditions for humanitarian aids.

## Figures and Tables

**Figure 1 foods-10-02077-f001:**
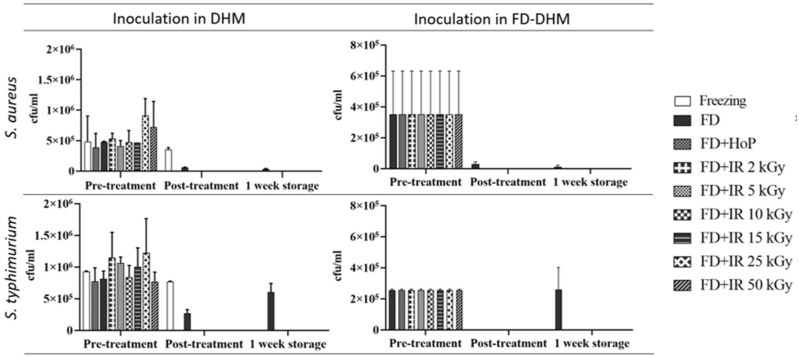
The effect of hybrid process (freeze-drying followed by gamma-irradiation) on the reduction of *S. aureus* and *S. typhimurium.* The column on the left shows the inoculation into liquid milk samples followed by freeze-drying and treatments. The column on the right shows inoculation on a freeze-dried DHM, prior to reconstitution, to simulate extrinsic contamination of FD-DHM powder. Values are the means ± SD of 2 technical replicates.

**Figure 2 foods-10-02077-f002:**
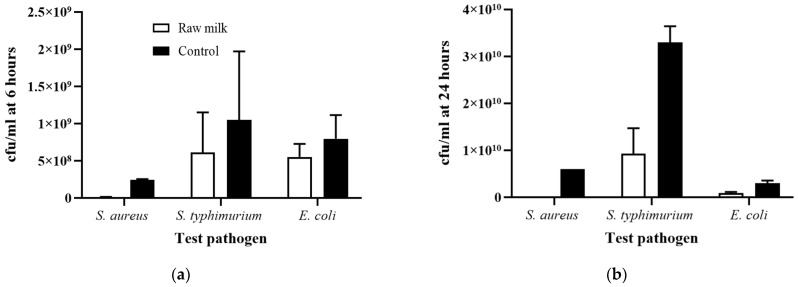
Bacterial growth of inoculants in raw DHM after 6 (**a**) and 24 h (**b**) incubation. Inoculants included *S. aureus, S. typhimurium*, and *E. coli* and were inoculated at 10^7^ cfu/mL. Control is inoculant growth in LB broth. Values are the means ± SD of 2 biological replicates.

**Figure 3 foods-10-02077-f003:**
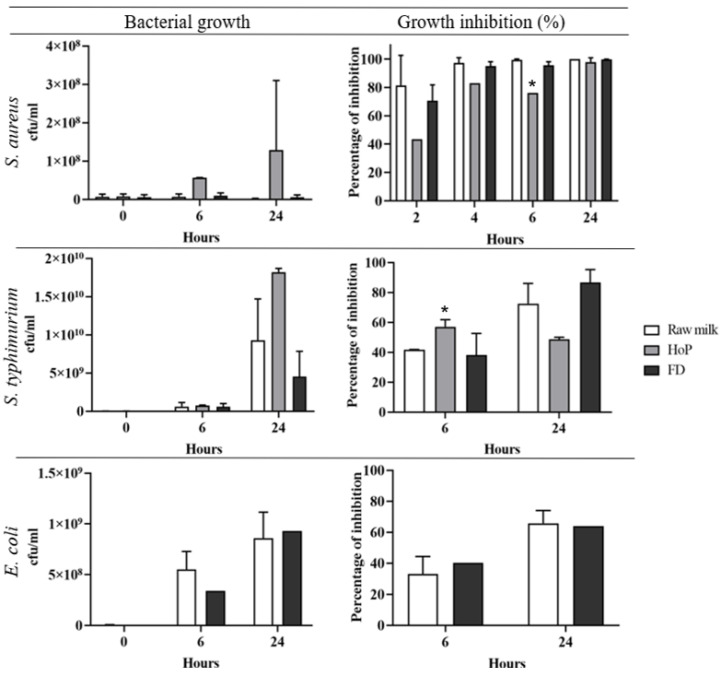
Bacterial growth and growth inhibition of inoculants in raw DHM and treated DHM. Treatments included freeze-drying (FD) and Holder pasteurisation (HoP). Inoculants included *S. aureus*, *E. coli*, and *S. typhimurium*, and 10^7^ cfu/mL was used for each inoculant. Values are the means ± SD of 2 biological replicates. * Significant change in antimicrobial activity compared to raw DHM, *p* ≤ 0.05.

**Figure 4 foods-10-02077-f004:**
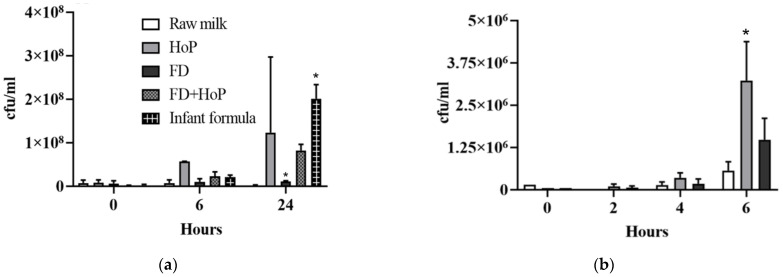
The microbial inhibitory effect of DHM and infant formula with respect to (**a**) *S. aureus* and (**b**) *E. coli* when processed by different techniques. Values are the means ± SD of 2 biological replicates. * Significant change in antimicrobial activity compared to raw DHM, *p ≤* 0.05.

**Figure 5 foods-10-02077-f005:**
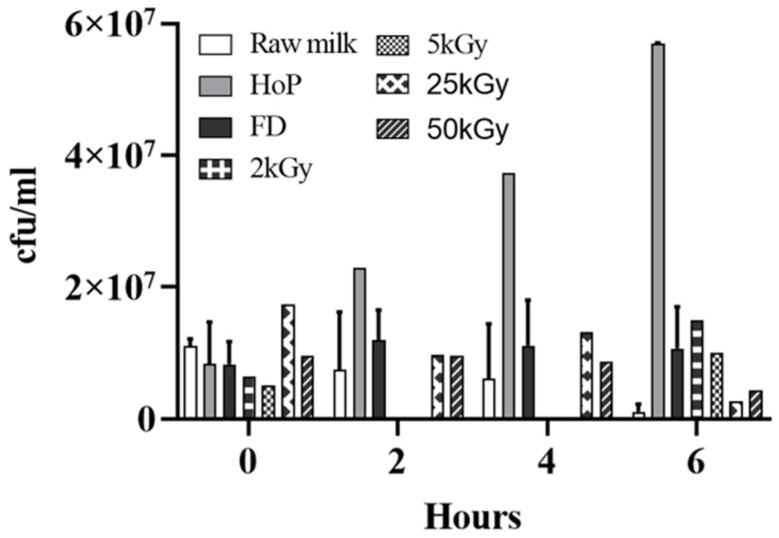
The comparison between the growth of *S. aureus* (inoculation dose 10^7^ cfu/mL) in DHM processed by different methods after 6 h. Raw DHM was compared to Holder pasteurised DHM, FD-DHM and gamma-irradiated FD-DHM at 2 and 6 h for 2 and 5 kGy, and at 2, 4, and 6 h for 25 and 50 kGy. Values are the means ± SD of 2 biological replicates for raw milk, HoP and FD samples, and 2 technical replicates for 2, 5, 25, and 50 kGy samples.

**Table 1 foods-10-02077-t001:** Percentage of live and dead *S. aureus* detected by flow cytometry. Live bacteria detected with SYTO9 and dead bacteria detected by DAPI. Measurements taken immediately after inoculation and 24 hours after incubation.

Sample	Percentage of Live Bacteria	
	Baseline	24 h
Raw milk	95.3	92.2
Freeze-dried	79.9	11.3
Holder pasteurised	99.0	93.9
Freeze-dried + Holder pasteurised	85.0	0.3
Infant formula	91.4	95.5
Positive control	99.2	99.2
Negative control	0.0	0.0

Positive control is live. *S. aureus* in saline, and negative control is dead *S. aureus* in saline. A single biological replicate was used for this experiment.

**Table 2 foods-10-02077-t002:** Percentage of growth inhibition of *S. aureus* in gamma irradiated FD-DHM samples. Growth inhibition measured by comparison to optimal growth in LB broth, and measured after inoculation at the time points 0, 6, and 24 hours. Values are the means ± SD of 2 biological replicates.

	Raw	HoP	FD	FD + HoP	Infant Formula	2 kGy	5–50 kGy
6	94.9 ± 0.8	67.7 ± 12.5	97.8 ± 1.0	90.7 ± 4.2	91.4 ± 2.0	99.7 ± 0.3	100 ± 0.0
24	100.0 ± 0.0	96.3 ± 1.4	99.9 ± 0.0	98.6 ± 0.2	96.7 ± 0.5	100 ± 0.0	100 ± 0.0

## Data Availability

The datasets generated for this study are available on request to the corresponding author.

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
