# Peer review of "The Effects of Thermal Pasteurisation, Freeze-Drying, and Gamma-Irradiation on the Antibacterial Properties of Donor Human Milk"

_foods, 2021, doi:10.3390/foods10092077_

Round 1

Reviewer 1 Report

This study aimed to determine the suitability of gamma-irradiation as a pathogen control process for freeze-dried donor human milk (FD-DHM). The antimicrobial activity of FD-DHM, holder pasteurized DHM and raw DHM, and infant formula was determined using S. aureus, Salmonella and E. coli. This study provide novel knowledge on new method to preserve immune component in DHM. 

Major corrections:

Statistical analysis is missing in the material and methods section and must be added and performed for the data interpretation (line 222-223).

Statistical analysis is missing in all figures and tables and must be added.

Please, specify the values presented in the figures (example: average +- SEM for n = X). 

Minor corrections.

Specify how many donors you used to make the pooled DHM (lines 112-113, p.3)

Author Response

Reviewer 1

  1. Statistical analysis is missing in the material and methods section and must be added and performed for the data interpretation (line 222-223).

This was added into the methods section (lines 224-226) as shown below:

Statistical analysis. Changes in antimicrobial activity in treated DHM samples were compared the antimicrobial activity in raw DHM and tested for significance using two-tailed t-tests in Excel (2016). GraphPad Prism 8.3.1 was used to create the figures.

  1. Statistical analysis is missing in all figures and tables and must be added.

Relevant information such as figures representing means +/- SD, technical replicates etc. have been added to the figure and table legends.

  1. Please, specify the values presented in the figures (example: average +- SEM for n = X). 

Completed – see above.

  1. Specify how many donors you used to make the pooled DHM (lines 112-113, p.3)

The number of donors (n= 5-6) has been added.

Reviewer 2 Report

This manuscript aims to evaluate the efficacy of a combination of freeze-drying and low-dose gamma irradiation as an alternative to thermal pasteurization. The results demonstrate that the improved retention of antimicrobial properties of irradiated FD-DHM compared to pasteurized DHM. Although the application at industrial level of this technology to powdered milk is not yet a commercial reality, it offers a novel idea of non-thermal processing.

The subject falls within the general scope of the journal and is a good attempt in the field of dairy processing based on prior technologies. The topic is well introduced, the procedure clearly described, and results are properly presented in tables and figures. However, appropriate modifications are still needed.

Line 177:

“Colony counts were performed at least in duplicate” means “each experiment was performed at least in duplicate”? They are different and please specify the number of samples in each experiment and identify them in the figure and table, such as “n=3”.

Line 181-183:

It is recommended to add the detailed calculation formula for the percentage of inhibition here to facilitate the elaboration and analysis in Figure 3.

Line 243:

Bacteria inoculated in FD-DHM before different treatments seem to have the same colony count, why it did not occur in DHM? Please explain the difference between the inoculation in DHM and inoculation in FD-DHM.

Line 252:

Do the two data here correspond to the pre-treatment data in the right column of Figure 1?

Line 286-287:

The ordinate scales are spaced only 0.5 log apart, which is quite small for microbial experiments. Although significant differences were mentioned in some results in the manuscript, they were not marked in the figure. Also, whether the error bar represents the standard deviation? This needs to be explicitly stated in figure legend. If so, the SD appears to be very large, so it is recommended that significant differences be added in all figures.

Line 308:

24 hours after HoP treatment, the percentage of inhibition for S.aureus reaches 97.6%, but the bacterial growth still seems to be more than 8 log. Please explain it. In addition, bacterial growth in Figure 3 seems to be barely discussed in the manuscript and is considered for deletion.

Line 317-319:

The growth inhibition mentioned here seems not to be directly visible in Figure 4a.

Line 323-324:

The words "A" and "B" in the legend should be the same case as the words "a" and "b" in figure 4.

Line 479-487:

The text in the supplementary Figure 5 is too blurred. It is recommended to enlarge the picture to help readers understand the manuscript.

Author Response

Reviewer 2

  1. Line 177:

“Colony counts were performed at least in duplicate” means “each experiment was performed at least in duplicate”? They are different and please specify the number of samples in each experiment and identify them in the figure and table, such as “n=3”.

When technical replicates and biological replicates have been performed, they are now included in line 212 (methods) and detailed in the figure legends.  

  1. Line 181-183:

It is recommended to add the detailed calculation formula for the percentage of inhibition here to facilitate the elaboration and analysis in Figure 3.

Formula for growth inhibition, which his shown in Figure 3 and Table 2 has been added and can be seen in line 186.

  1. Line 243:

Bacteria inoculated in FD-DHM before different treatments seem to have the same colony count, why it did not occur in DHM? Please explain the difference between the inoculation in DHM and inoculation in FD-DHM.

When DHM was inoculated with test pathogens, it was possible to test each sample for colony counts. In contrast, when FD-DHM was inoculated with test pathogens (to measure the efficacy of irradiation for pathogen control), it was not possible to test all samples as they needed to be irradiated before reconstitution. Therefore, two samples were reconstituted and the colonies present were enumerated. This value was then assumed to represent how much of the test pathogen was able to survive in DHM powder when not reconstituted. Hence, all the values for the colony counts are equivalent for FD-DHM in Figure 1. This is now clarified in line 266 “These values were presumed to be representative of all inoculated FD-DHM samples”

  1. Line 252:

Do the two data here correspond to the pre-treatment data in the right column of Figure 1?

Yes, that is correct. Two sentences (lines 253-256) have been added to the figure legend in Figure 1 to clarify what the two different columns are: “The column on the left shows inoculation into liquid milk samples followed by freeze-drying and treatments. The column on the right shows inoculation on a freeze-dried DHM, prior to reconstitution, to simulate extrinsic contamination of FD-DHM powder. Values are the means ± SD of 2 technical replicates.”

  1. Line 286-287:

The ordinate scales are spaced only 0.5 log apart, which is quite small for microbial experiments. Although significant differences were mentioned in some results in the manuscript, they were not marked in the figure. Also, whether the error bar represents the standard deviation? This needs to be explicitly stated in figure legend. If so, the SD appears to be very large, so it is recommended that significant differences be added in all figures.

Where statistical analyses were performed, significant differences have now been marked in the figures (see Figures 3 and 4). An explanation of the values and error bars have now been added to the figure legends for all Figures and Tables.

  1. Line 308:

24 hours after HoP treatment, the percentage of inhibition for S.aureus reaches 97.6%, but the bacterial growth still seems to be more than 8 log. Please explain it. In addition, bacterial growth in Figure 3 seems to be barely discussed in the manuscript and is considered for deletion.

Growth inhibition of DHM and treated samples were compared to a positive control, which was bacterial growth in LB broth. There was only a 2.4 % decrease in HoP DHM compared to DHM using bacterial growth in LB broth as a standard. But there was a 6-fold increase in bacterial growth in HoP DHM compared to DHM. A sentence was added (Line 311) to try to clarify this matter. “While this is a small decrease when compared to bacterial growth in the positive control (LB broth), 2.4 % is equal to a 6-fold increase of bacterial growth (Figure 3). “

  1. Line 317-319:

The growth inhibition mentioned here seems not to be directly visible in Figure 4a.

This refers to Figure 3 (line 316).

  1. Line 323-324:

The words "A" and "B" in the legend should be the same case as the words "a" and "b" in figure 4.

These have now been changed to match Figure 4 as recommended.

  1. Line 479-487:

The text in the supplementary Figure 5 is too blurred. It is recommended to enlarge the picture to help readers understand the manuscript.

The text has been enlarged so it can now be clearly read.

Round 2

Reviewer 1 Report

The authors corrected the concerns/comments.